# Water Ecosystems Tool (WET) 1.0 – a new generation of flexible aquatic ecosystem model

Nicolas Azaña Schnedler-Meyer[1], Tobias Kuhlmann Andersen[2,3], Fenjuan Rose Schmidt Hu[2], Karsten Bolding[2,3,4], Anders Nielsen[2,3] and Dennis Trolle[2,3]

[1] National Institute of Aquatic Resources, Technical University of Denmark (DTU), Silkeborg, Denmark.

[2] Aarhus University, Department of Ecoscience, Vejlsøvej 25, 8600 Silkeborg, Denmark.

[3] Sino-Danish Centre for Education and Research, University of Chinese Academy of Sciences, Beijing, China.

[4] Bolding & Bruggeman ApS, Strandgyden 25, 5466 Asperup, Denmark.

*Correspondence to*: Nicolas A. Schnedler-Meyer (niaz@aqua.dtu.dk)

**Abstract.** We present the Water Ecosystems Tool (WET) - a new generation of an open source, highly customizable aquatic ecosystem model. WET is a completely modularized aquatic ecosystem model, developed in the syntax of the Framework for Aquatic Biogeochemical Models (FABM), which enables coupling to multiple physical models ranging from zero to three dimensions, and is based on the FABM-PCLake model. The WET model has been extensively modularized, empowering users with flexibility of food web configurations, and incorporates model features from other state-of-the-art models, with new options for nitrogen fixation and vertical migration. With the new structure, features and flexible customization options, WET is suitable in a wide range of aquatic ecosystem applications. We demonstrate these new features and their impacts on model behavior for a temperate lake for which a model calibration of the FABM-PCLake model was previously published, and discuss the benefits of the new model.

## 1 Introduction

The study and management of aquatic ecosystems have benefitted widely from the ongoing development of various numerical model approaches to a host of ecological questions (Soares and Calijuri, 2021). As the field matures, new and superior approaches and descriptions of individual ecological processes are formulated and improved upon, and management tools must continuously be updated to reflect the current state-of-the-art. However, rather than building new models from scratch, and thus 're-inventing the wheel' over and over (Trolle et al., 2012), another way forward is to consolidate new descriptions of ecological processes into a few proven and well-established biogeochemical models, thereby improving their applicability for management and for the study of ecosystem-wide responses to environmental stressors. There is therefore a call for flexible and configurable models that contain various optional features, allowing them to be tailored to specific uses, without changing the model code or making a new version.

Among the most widely used lake ecosystem models in the world (Mooij et al., 2010; Trolle et al., 2012), the PCLake model was originally developed for the shallow Dutch lake Loosdrecht in the early 90s, under the name PCLoos (Janse and Aldenberg,

1990). Extended and renamed, this 0D model has since been used to analyze regime shifts and eutrophication responses in fully mixed temperate shallow lakes (Janse, 1997, 2005; Janse and van Liere, 1995; Mooij et al., 2010 and references herein; Rolighed et al., 2016). Mostly based on mechanistic process descriptions, the model is relatively complex with ~100 state variables, covering dry weight, phosphorous, nitrogen and silica dynamics in both the water column and the sediment and accounting for inorganic nutrients, detritus, and a fixed food web (see Janse, 2005 for a full description). The original model has been made available in several formats (see Mooij et al., 2010 for an excellent summary), and has since been independently adapted, reconfigured and extended by various authors into several parallel versions tailored to specific applications and physical setups. These include a static neural network metamodel for estimating critical P loadings, a subtropical version, and various 1D, 2D and 3D versions (see Mooij et al., 2010, and references herein). Most recently, a modified version of the original 0D model, extended with an optional hypolimnion layer was published (Janssen et al., 2019). The model has proved useful in a range of case studies, exploring different management and climate scenarios (e.g. Janssen et al., 2015; Mooij et al., 2010; Rolighed et al., 2016), and given the fact that it is open source, it has become a starting point for the development of more specialized models, as is apparent from the numerous versions that have arisen over the last decades. Such parallel development is a sign of the general success of the original model, but is unfortunate, as it risks multiple 're-inventions of the wheel' (Trolle et al., 2012) along the way. Even worse, useful updates to some versions of the model that could benefit all versions risk being lost.

A large step towards a more flexible and generally applicable version of PCLake was taken with the development of the FABM-PCLake model by Hu et al. (2016), who recoded the original model into the syntax of the Framework for Aquatic Biogeochemical Models (FABM, Bruggeman and Bolding, 2014) and modified the basic formulations to allow the study of spatial dynamics within deeper, stratifying aquatic environments, thus opening up the applicability to a much broader range of aquatic systems worldwide. The Framework for Aquatic Biogeochemical Models (FABM), allows coupling a biogeochemical model to a wide variety of hydrodynamic models, in 0-, 1-, 2- or 3D (Bruggeman and Bolding, 2014), without changing any model code, and encourages and supports modularization of ecosystem models. However, though FABM-PCLake is much more flexible with regards to spatial setups and type of modelled system, and also includes additional species of organic matter, it is still essentially identical to the original model in its biological descriptions, and has inherited the ecological rigidity and limitations of the original model, with its focus on shallow eutrophic lake ecosystems. As an example, many organisms employ vertical movement (VM) as a means to exploit vertical gradients in e.g. nutrient, oxygen or light availability (e.g. Dini and Carpenter, 1992; Mehner, 2012; Olli, 1999). Existing model variants in the PCLake family were primarily developed for shallow lake applications, and do therefore not consider the ecological ramifications of lake stratification or the movement of organisms in the vertical. While the FABM-PCLake model have been applied to deeper, stratified lake systems in a 1D setup (e.g. Allan, 2018; Chen et al., 2020), its biological descriptions and structure has inherited some limitations in how it deals with spatial heterogeneity from its 0D predecessors. Examples of this includes the fact that FABM-PCLake has no exchange of fish between model depth layers, even when depth resolution is fine (e.g. layers being only a few centimeters thick), and

that movement of plankton elements is limited to passive advection and a constant sinking or flotation velocity. These limitations might be acceptable in shallow environments or 0D applications, but quickly become untenable in deeper systems. Here, we present a complete restructuring and -coding of the FABM-PCLake model that adds both flexibility as well as new features to the model. To avoid conflation with the increasing number of PCLake versions available, we have decided to present this new model under its own distinct name, the Water Ecosystem Tool (WET). So far, the name Water Ecosystem Tool (WET) has been associated to the QGIS plugin developed by Nielsen et al. (2017) to setup, configure and run the coupled GOTM-FABM-PCLake model complex. With this paper, we redefine what WET is, and present it as a new generation of an aquatic ecosystem model, originating from the PCLake model, specifically the version by Hu et al. (2016).

In the following sections, we present the suite of new features which have been added to the PCLake framework, together with example dynamics from its first application to a lake ecosystem – a temperate Danish lake. The new features constitute a complete modularization of the model code, the inclusion of vertical migration algorithms and the addition of a nitrogen fixation option to the phytoplankton module. A plugin (now called QWET) for the GIS software QGIS has also been developed (Nielsen et al., 2021), which provides a graphical user interface to configure and run WET in a user-friendly workflow in conjunction with the 1D hydrodynamic model GOTM, and allows (but does not require) linking GOTM-WET to the SWAT and SWAT+ (Soil & Water Assessment Tool) watershed model (Arnold et al., 1998). The plugin can be found on the WET website ([http://wet.au.dk](http://wet.au.dk)), along with user guides instructing the user on how to download and set up both WET and QWET.

## 2 Model description

Like its predecessor FABM-PCLake, WET can describe interactions between multiple trophic levels and abiotic nutrient dynamics in both the water column and the sediment. The model accounts for the dynamics of dry weight, nitrogen, phosphorous, silica and oxygen, and features bottom-shear-dependent resuspension, as well as two different light-limitation functions for phytoplankton. WET is also implemented within the FABM framework, allowing the model to be coupled to various physical driver models, e.g. GOTM (1D, Burchard et al., 1999) or GETM (3D, e.g. Stips et al., 2004), without changing any of the model code. Within the FABM framework, the physical model takes care of updating and iterating the model state variables forward in time, and the sole responsibility of the WET code is to provide local source and sink terms for its state variables as well as feedback to physical variables such as light or bottom shear stress (Bruggeman and Bolding, 2014).

Here we concentrate on descriptions of the new features and all changes that separate WET from its parent model. We refer to Hu et al. (2016) and Janse (2005; 1992; 1995) for a detailed description of the basic equations governing the biogeochemical processes and food web dynamics, since these are unchanged, even though the model code has been rewritten and reorganized. A complete list of parameters related to the new features with options and default values can be seen in Table 1.

## 2.1 Modularization of the food web

A major drawback of the FABM-PCLake model is that it retains the rigid food web structure inherited from the original PCLake model and can only run in a fixed food web configuration, with fixed, preordained interactions between food web components. In contrast, WET has been designed to take full advantage of FABM, by being fully modularized. This modularization enables the user to set up an arbitrary number of types of any food web element (e.g. multiple phytoplankton types) within a simulation, or to remove it altogether (e.g. no fish). Thus, it is possible to customize the model to a desired level of complexity with the aim of addressing a specific study system or research question, and without changing any of the model code. Increasing or decreasing simulated food web complexity as the situation requires or data allows is done by simply adding or removing a food web module instance from a single configuration file (fabm.yaml, see Fig. 3). As an example, to add a new zooplankton species to an already existing model, one could first copy the lines for an existing zooplankton species in the configuration file, and then change the name of the copy instance. Secondly one would go through the *couplings* and *parameters* sections in the new zooplankton instance, modifying these to fit the desired organism. Finally, one would modify the instances of any predators to include the new zooplankton instance in their diets. Thus, adding or subtrackting instances to a model setup is relatively easy, and testing for the optimal food web configuration in a specific case is possible, if not usually feasible, by calibrating several different module setups and comparing their performance.

The code base of the WET model consists of eleven FORTRAN files. Six of these are required files, of which two handle model initialization and shared functions, and four constitute a basic chassis of required modules, handling microbial, chemical and physical processes in the water column and upper sediment (see Fig. 2, 'fixed modules'). Besides these core parts, the model consists of five optional modules representing different food web component types (phytoplankton, rooted macrophytes, zooplankton, zoobenthos, and fish, see Fig. 2, 'food web modules'). For all WET modules, fabm.yaml testcase setup files with default parameters can be found in the testcase folder of the source file repository.

### 2.1.1 Primary producers

The WET phytoplankton module is developed with a high degree of flexibility in mind, and contains options to allow it to represent all main phytoplankton groups. These constitute optional dependence on silica for growth (e.g. for modelling diatoms), the option to allow phytoplankton to fix atmospheric nitrogen (e.g. cyanobacteria), and the optional ability to migrate vertically in response to ambient light and nutrient availability (various phytoplankton groups). Both nitrogen fixation and vertical movement algorithms are new features of WET, described in the following sections (Sect. 2.2 and 2.3, respectively). The option of silica dependence is turned on by setting the a parameter in the configuration file (fabm.yaml, see Fig. 3 and Table 1), and its formulation is otherwise identical to the one used in the PCLake model family (Hu et al., 2016; Janse, 2005; Janse and van Liere, 1995).

In WET, multiple types of rooted macrophytes can be included. Depending on the requirements of the modeler, all or some of the macrophyte instances can be set up to share a common carrying capacity, by changing a parameter in all competing macrophyte instances, and pointing to the relevant macrophyte instances in the configuration file. Aside from this option and

the general modularization, the formulation of the macrophyte module is identical to FABM-PCLake (Hu et al., 2016; Janse, 2005; Janse and van Liere, 1995).

### 2.1.2 Heterotrophic modules

In accordance with the modularization of WET, the original feeding formulations (see Janse, 2005) of heterotrophic modules

– zooplankton, zoobenthos, and fish – have been adapted to be flexible, allowing predators to feed on multiple prey (including other instances of their own base module). Mixed diets are set up in the configuration file, where each prey is pointed to, and a preference factor (typically between 0 and 1) is specified, following e.g. Fasham et al. (1990). In addition, consumption of particulate organic matter (POM) is likewise an option for the invertebrate modules, i.e. zooplankton and zoobenthos.

For fish, foraging is separated into three foraging modes, planktivory, benthivory, and piscivory. These foraging modes are

assumed separate in time and/or space, such that each take up a fraction (*fFishZoo*, *fFishBen* and *fFishPisc*, which must sum up to 1) of the total foraging effort of the fish population. For each mode, several prey types can be present, each with their own preference factor as for zooplankton and –benthos. Saturation functions are calculated for each foraging mode separately, using a Monod-type formulation, e.g. for piscivory:

$$aDSatFiPisc = \frac{\sum_{i=1}^{nPISC}(sDPisc_i \times PISCpref_i)}{hDPiscFi + \sum_{i=1}^{nPISC}(sDPisc_i \times PISCpref_i)},$$ (1)

where *aDSatFiPisc* is the current saturation level for piscivorous feeding, *nPISC* is the number of fish prey types, *SDPisc$_i$* is the biomass of fish prey *i*, *PISCpref$_i$* is the preference factor for fish prey *i*, and *hDPiscFi* is the half-saturation constant for piscivorous feeding. The amount of assimilated biomass from each foraging mode (here *tDAssFiPisc*, again using piscivory

as an example) is then calculated as:

$$tDAssFiPisc = aDSatFiPisc \times sDFi \times (kDAssFi \times aFunVegAss \times uFunTmFish) \times fFishPisc,$$ (2)

where *sDFi* is the biomass of the predator, *kDAssFi* is the maximum assimilation rate of fish at 20°C, *aFunVegAss* is the

(optional) dependency of piscivory on macrophyte biomass (Janse, 2005; Janse and van Liere, 1995), and *uFunTmFish* is the temperature correction on fish vital rates (calculated internally in WET). The contributions from the three foraging modes are then summed for the purpose of calculating total assimilation.

### 2.1.3 Linking fish instances into a pseudo stage structure

WET describes all populations in terms of biomasses, and does not explicitly consider population or age structure of any organism. For fish however, the link between juvenile (zooplanktivorous) and adult (benthivore) fish present in the PCLake model family (Hu et al., 2016; Janse, 2005; Janse and van Liere, 1995), has been generalized to the fish module. Thus, in WET, instances of the fish module can be linked through 'aging' or 'reproduction', where a fixed proportion of biomass is transferred from one instance to another on a fixed date. For both aging and reproduction, this is set up in the configuration

file, by setting the *qStageOpt* parameter, pointing to the recipient fish instance(s), and providing parameters for the date of transfer and the proportion of biomass transferred. In this way, a population of fish can be separated into a stage structure, containing two, three or more stages, with individual parameterizations, diets and predators. Note, however, that this implementation is not truly a stage-structured model, as each instance in the structure can in principle persist indefinitely, regardless of the state of the others.


### 2.2 Phytoplankton nitrogen fixation

Depending on the external nutrient inputs, nitrogen fixation by cyanobacteria can be an influential process in freshwater ecosystems (e.g. Paerl et al., 2016). Advancing from the PCLake model family (Hu et al., 2016; Janse, 2005; Janse and van Liere, 1995), WET features the possibility to simulate nitrogen fixation, which can be turned on or off by the user for each

individual phytoplankton instance. This is done by setting the *lNfix* parameter in the configuration file (see Table 1), and supplying two additional parameters; the maximum fixation rate (*cNFixMax*, mgN mgDW$^{-1}$ d$^{-1}$), and the maximum realized fraction of the growth rate at maximum nitrogen fixation rate (*fMuNfix*, dimensionless). By default, total nutrient limitation for phytoplankton growth is governed by Liebig's Law of the Minimum, and is by default calculated as:

$$aNutLim = \min \begin{cases} aPLim \\ aNLim \\ aSiLim \end{cases} \qquad\qquad (3)$$

where *aNutLim* is the overall nutrient limitation, and *aPLim*, *aNLim* and *aSiLim* are the Droop functions for phosphorous, nitrogen and silica growth limitation, respectively. In the case of nitrogen fixation being turned on, nitrogen uptake rate is assumed to never limit phytoplankton growth, and consequently, total nutrient limitation is:


$$aNutLim = \min \begin{cases} aPLim \\ aSiLim \end{cases} \qquad\qquad (4)$$

or simply *aPLim*, in the absence of silica uptake. This independence of internal nutrient concentration for growth dynamics is balanced by a growth rate reduction due to allocation of energy to nitrogen fixation:


$$aNFixLim = fMuNFix + aNLim(1 - fMuNFix) \qquad (5)$$

Note that this formulation assumes that nitrogen fixation takes place whenever the phytoplankton is less than nitrogen replete (and in spite of other possible limiting nutrients) and that phytoplankton only fixes what nitrogen it cannot uptake through

mineral absorption. The final nitrogen fixation rate is calculated as:

$$aNFix = cNFixMax(1 - aNLim) \qquad (6)$$

and has units of mg N mg$^{-1}$ DW d$^{-1}$. This formulation of the nitrogen fixation dynamics in WET is adapted from the CAEDYM

model (Hamilton and Schladow, 1997; Hipsey et al., 2005), while this general formulation of nitrogen fixation is common in phytoplankton models (see e.g. Inomura et al., 2020 and references therein).

### 2.3 Vertical movement algorithms

In WET, all pelagic food web modules (i.e. phytoplankton, zooplankton and fish) now have several options for vertical

migration. The purpose of these options is to 1) Add flexibility to the types of environment that can be modelled with WET, and 2) To increase model accuracy and applicability by providing more realistic dynamics of all food web elements, in all types of aquatic systems. The various options for vertical migration are presented for each model element below. These options can be configured individually for each instance at runtime by setting the *qTrans* parameter, and any necessary additional option-specific parameters.


### 2.3.1 Phytoplankton module

The WET phytoplankton module contains four modes of vertical movement behavior: Passive transport (no active movement), passive transport plus active chemotaxis (for nutrients), passive transport plus active phototaxis, and passive transport plus combined photo- and chemotaxis (see Table 1 for details on options and parameterization). The VM functions of phytoplankton

described below are all based on Ross and Sharples (2007).

The first vertical movement option is passive transport, identical to what is currently the only available mode in FABM-PClake, where phytoplankton only move vertically as a result of passive advection by the host physical model as well as through a fixed sinking rate (positive, negative or neutral buoyancy). The fixed sinking rate is specified through the *cVSet* parameter, and is negative in case of sinking.

In addition to passive advection, when chemotaxis is turned on, phytoplankton will swim downwards with constant swimming speed, whenever nutrient limitation growth coefficient decreases below a threshold value. Thus, phytoplankton in WET operate under the assumption that nutrient concentrations are always higher at greater depth.

When phototaxis is turned on, phytoplankton will swim upwards with fixed speed, whenever ambient light levels surpass a light threshold value.

The fourth phytoplankton vertical movement option combines chemotaxis and phototaxis, such that chemotaxis takes precedence over phototaxis. Thus, phytoplankton will swim down in nutrient deplete situations, and up when the cell is nutrient replete, provided that ambient light levels surpass the threshold value.

### 2.3.2 Zooplankton & Fish modules

Regular vertical movements between different depths are a common behavior in both fish and zooplankton populations, especially in deeper systems. Such movement behavior is expressed for a variety of reasons, including avoidance of hypoxic regions, predator avoidance, and bioenergetic exploitation of gradients in temperature or food availability (Dini and Carpenter, 1992; Dodson, 1990; Lambert, 1989; Mehner, 2012).

An inherent limitation to the FABM framework is that modules are limited in the amount of information they receive about
conditions outside the current model cell, e.g. food availability at other depths. Thus, modelled motile organisms are limited to making 'decisions' about movement based on either local conditions or predictable environmental gradients. Due to this limitation, directional vertical movement of zooplankton and fish in WET are restricted to being in response to hypoxia, ambient light levels, or both. The WET zooplankton and fish modules contain four modes of vertical migration behavior: No transport (no transport between layers, even advection), passive transport (no active transport), hypoxia avoidance, and light-
based diel vertical migration combined with hypoxia avoidance. Of these, the no transport option turns off all exchange between model layers (only relevant in >0D applications), and is mainly a debugging or analytical tool.

As for phytoplankton, when vertical movement is set to passive transport, fish and zooplankton are passively advected by the physical model.

Hypoxia avoidance restricts the habitat domain to exclude anoxic parts of the water column, a ubiquitous response to hypoxia
among zooplankton and fish (Ekau et al., 2010; Vanderploeg et al., 2009). When hypoxia avoidance is turned on for a WET module, fish or zooplankton swim upwards whenever the ambient oxygen concentration falls below the critical threshold in the current model cell.

Zooplankton and fish may employ diel vertical migration for a number of reasons (Dini and Carpenter, 1992; Dodson, 1990; Lambert, 1989; Mehner, 2012; Sainmont et al., 2013), however ambient light levels is often the proximate trigger for this
behavior. Under this setting, in addition to passive advection and hypoxia avoidance, fish or zooplankton will swim downwards, whenever light exceeds a maximum level, and upwards whenever light decreases below the lower light threshold.

Either the downwards or upwards portions of this movement can be turned off, by setting the maximum (minimum) threshold to a very high (negative) value.

## 3 WET testcase – Lake Bryrup

To illustrate some of the new features in the WET model, we applied the GOTM-WET model for Lake Bryrup, for which a calibration of the FABM-PCLake model (coupled to the lake version of the 1D hydrodynamic model GOTM, Burchard et al., 1999) was previously published by Chen et al. (2020). Here we have adapted, recalibrated and validated this setup using WET, following the methodology and approaches of Chen et al. (2020), and use the results to illustrate some of the new features of WET, and how these can impact the model behavior.

### 3.1 Study site and model configuration

The shallow Lake Bryrup is located in the Central Region of Denmark (56.02° N, 9.53° E), and has a surface area of 37 hectares, a mean depth of 4.6 meters, and a maximum depth of 9 meters. The lake stratifies temporarily during summer, and has a water retention time of 2-3 months. The catchment area is 49.9 square kilometers and heavily farmed, and the lake receives large amounts of nutrients from agricultural and urban drainage (Johansson et al., 2019). Consequently, the lake is eutrophic although management measures have been effective in reducing average total nitrogen and phosphorous concentrations throughout the last decades (see Chen et al., 2020, for a more thorough description of Lake Bryrup).

As the chosen physical driver model for the Lake Bryrup test case, we used the Aarhus University fork of lake branch GOTM (version 5.2.2-au, available at https://gitlab.com/WET/gotm), configured with a maximum depth of 9.0 and 18 vertical layers (i.e. vertical grid size of 0.5 m). In order produce high-resolution output for the present figures, the model was also run with a 200 layer resolution. The setup used a lake-specific hypsograph (i.e. the relation between depth and horizontal area) a to capture lake sediment-water column interactions at all depths, by effectively splitting the bottom between model layers, such that each model layer in the 1D setup has an attached bottom layer. Interactions between the water column of a layer, its attached bottom, and the water column layer below is governed by the hypsograph, which specifies the fraction of the bottom area to total layer area. The setup therefore in some aspects functions as a pseudo-2D setup. Each model layer thus included a sediment layer of 10 cm, similar to the sediment compartment of the PCLake model. See Andersen et al. (2020) and Hu et al. (2016) for detailed descriptions of these aspects of the model. To simulate lake ice thickness and cover, implementation in GOTM of the ice module from MyLake (Saloranta and Andersen, 2007) was enabled. In WET, the food web comprised three phytoplankton groups (*diatoms*, *cyanobacteria* and *other algae*), rooted, submerged macrophytes, two zooplankton groups (*Daphnia* and *other zooplankton*), detritivorous macrozoobenthos, juvenile zooplanktivorous and adult zoobenthivorous fish and piscivorous fish (Fig. 1). We applied the European ECMWF ERA5 dataset (Hersbach et al., 2018) at an hourly resolution on air temperature (° C), air pressure (hPa), dew-point temperature (° C), cloud cover (%) and wind speed components (m/s) in the north-south

and west-east direction as meterological forcing for GOTM. Monthly averages of water inflow ($m^3/s$) and nutrient concentrations ($NO_3$, $NH_4$, $PO_4$ and particulate organic matter of nitrogen and phosphorous, mg/L) from the National Monitoring and Assessment Program for the Aquatic and Terrestrial Environment in Denmark (NOVANA) (Kronvang et al., 1993; Lauridsen et al., 2007) were used as boundary conditions (see Chen et al., 2020, for details), applied in the topmost model layer (both in- and outflow). The model was executed with an hourly time step and daily (midday) output of model results, except for the high resolution runs, which ran with a ten minute time step and output.

## 3.2 Model calibration and validation

The GOTM-WET model for Lake Bryrup was calibrated following the calibration procedure described in Chen et al. (2020), which applied the auto-calibration tool parsac (Bruggeman and Bolding, 2020), with 6 spin-up years to initialise biogeochemical nutrient pools, a calibration period of 8 years (1996-2004) and a validation period of 2 years (2004-2006). Model results were compared against monthly to semi-monthly data for water temperature ($^o$ C), dissolved oxygen (mg/L), inorganic nutrient concentrations ($NO_3$, $NH_4$ and $PO_4$, mg/L), total nitrogen and phosphorous concentrations (TN and TP, mg/L) and chlorophyll a concentrations (chl. a, µg/L) obtained from the NOVANA program available at www.miljoportal.dk. Spatial resolution of in-lake dataset for calibration and validation varied with 1-12 measurements per sample date across the water column for water temperature and DO (median of 7 measurements per date) and 1-3 samples per date for water nutrient concentrations (median of 2). All chl a. concentrations were sampled in the surface (between 0-3 m depth) once per sample date. We evaluated model performance by computing root-mean-square-error (RMSE) and the coefficient of determination ($R^2$) for the daily output of each state variable. Model results have been processed and visualized in Python with the packages xarray version 0.15.1 (Hoyer and Hamman, 2017), matplotlib version 3.1.2 (Hunter, 2007), and seaborn version 0.11.1 (Waskom, 2021), as well as with the open-source Python program PyNcView (available via The Python Package Index, *pip install pyncview*).

## 4 Results of the testcase with example dynamics of new features

The new WET version of Lake Bryrup model differs from the FABM-PCLake version in a few areas, first and foremost by taking advantage of the new options for vertical mobility for motile cyanobacteria and zooplankton, and by allowing dispersion of fish between layers (see Sect. 2.3). In addition, the new modularization have allowed us try out an alternate food web configuration, namely the differentiation of zooplankton into two boxes (mesozooplankton and daphnia), although in this instance we are limited by the availability of data for calibration.

The new model yields comparable or slightly improved overall performance metrics to the ones obtained by Chen et al. (2020), with slightly lower or equal performance statistics for oxygen and nitrogen state variables, but improved performance for phosphorous, chlorophyll a, and temperature (see Table 2).

Figures 3-5 illustrate select dynamics from the Lake Bryrup WET model. For these figures, the calibrated model was rerun at a higher resolution in time and space, relative to the base calibration, in order to increase visual detail of spatial and temporal dynamics. The obtained high-resolution results were similar to the dynamics of model runs with lower resolution.

## 4.1 Phytoplankton modularity

As in Chen et al. (2020), the WET recalibrated Lake Bryrup model contains three phytoplankton groups, diatoms, green algae and cyanobacteria (Fig. 3). The general seasonal phytoplankton succession in Lake Bryrup involves a spring diatom bloom, an early summer bloom of green algae, and a cyanobacteria bloom late summer, or when the water column stratifies for a prolonged period. The central panels of Fig. 3 illustrates the separation of multiple phytoplankton groups into separate versions of the same general module, making it easy to add, switch out or remove phytoplankton groups from the model, and to parameterize these individually. The chosen set of phytoplankton categories reflects the available data, and matches the previous FABM-PCLake version of the model.

## 4.2 Phytoplankton nitrogen fixation

To illustrate the interplay/dynamics between nutrient limitation, nitrogen fixation and growth of cyanobacteria with and without nitrogen fixation ability, we included an additional cyanobacteria group with identical parameters to the calibrated cyanobacteria group besides turned on N fixation with default values (*lNfix* = true and *fMuNFix* = 0.9). Although phytoplankton groups in Lake Bryrup are P limited in the period where cyanobacteria blooms frequently occur (approx. mid-June to mid-August with simulated P limitation between 0.8 to 0.5), N-fixing cyanobacteria dominated the community in contrast to the non N-fixing cyanobacteria (Fig. 4A and B). Both cyanobacteria groups were N limited in Spring, which allowed N-fixing cyanobacteria to get an advantage by decreasing N limitation via N-fixation (Fig. 4C) and increase biomass concentrations, thereby having a head start for the period with low mixing and warmer surface waters. As expected, N-fixation rates increased with increased N limitation and N-fix. Cyanobacteria biomass concentration (Figure 4C, secondary y-axis). The relatively low N limitation experienced by the N-fixing cyanobacteria group (N limitation factor between 0.95 to 1.0) resulted a low growth rate penalty during periods with N-fixation between 0 to 1.5%. In scenarios with altered external nutrient loads to switch from a P-limited to a N-limited system in late summer to fall, N-fixing cyanobacteria still dominate the phytoplankton community in late summer with N-fixation rates now significantly increased (for instance 20-fold in one scenario) as the cyanobacteria are now more N limited.

## 4.3 Phytoplankton vertical mobility

Figure 5 illustrates the new options for modelling vertical movement for phytoplankton in WET, panel A illustrating water column temperature, panel B illustrating overall nutrient conditions (for cyanobacteria) in the water column, and panels C-F illustrating the dynamics associated with each of the four vertical movement settings (here only for cyanobacteria). Of these, panel C corresponds to the old FABM-PCLake condition (Hu et al., 2016). Much of the seasonal succession of phytoplankton, and especially the shift from denser-than-water and non-motile phytoplankton (pre-July dynamics of Fig. 5C-F, see also Fig. 3A and B) to a mid-to-late-summer cyanobacterial bloom are driven mainly by the presence/absence and location of lake stratification (as indicated by the temperature gradients in Fig. 5A). . The early part of the growing season are dominated by relatively fast-sinking non-motile diatoms and green algae, which have *qTrans* set to 1 (*passive advection* and *fixed sinking rate*, see Sect. 2.3.1), in the configuration file (Fig. 3).

With the formation of a late summer stable stratification, relatively buoyant or motile cyanobacteria (Fig. 3C, Fig. 5C-F) become dominant, in part due to their ability to stay in the epilimnion. Even under the passive transport setting (Fig. 5C), cyanobacteria are able to sustain a bloom under stratified conditions, due to their slow sinking rate (cVSet = -0.022 m d$^{-1}$). However, when phototaxis is turned on (Fig. 5E), higher biomasses are reached by the cyanobacteria. When nutrient taxis without phototaxis is turned on (Fig. 5D), cyanobacteria aggregate at the bottom of the mixed layer when nutrients at the surface are very scarce (Fig. 5B). Under the combined nutrient and phototaxis setting (Fig. 5F, cyanobacteria aggregate around the bottom of the thermocline, forming a deep chlorophyll maximum (DCM), where nutrients are more available, and cyanobacteria are less nutrient limited (i.e. higher simulated nutrient limitation factor), but are also able to exploit higher light intensities at the surface, resulting in higher biomasses, compared with nutrient taxis alone..

## 4.4 Zooplankton vertical mobility

In WET, heterotrophic pelagic modules such as fish and zooplankton can exhibit vertical movement, in the forms of hypoxia avoidance behavior and light-triggered diel vertical migration (see Sect. 2.3.2). Figure 6 illustrates the three options for zooplankton and fish vertical mobility behavior in WET. Triggered by the up- and downwards migration of the hypoxic deep zone (Fig. 6D), and the daily light cycle (Fig. 6F), the Daphnia in Fig. 6A conducts diel vertical migrations when *qTrans* is set to 3. The active movement of zooplankton is modulated by advection (Fig. 6E), which diffuses part of the migrating zooplankton during days of high turbulent mixing (e.g. on the 3rd and 7th of July in Fig. 6). In panel B of Fig. 6 (*qTrans* = 2), upwards swimming is triggered when hypoxic conditions extends up into the water column, which does not happen when vertical movement of Daphnia is limited to passive advection (panel C of Fig. 6, *qTrans* = 1).

## 5 Discussion and conclusion

**5.1 Observations on the performance of the new vertical movement and nitrogen fixation features.** Model configuration and complexity is often constrained by data availability. Symptomatically, although Lake Bryrup is included in the Danish long-term ecological monitoring program (NOVANA), there is no data against which to validate, for instance, spatial distributions of higher organisms. Here, we have been mostly concerned with demonstrating the features of WET, however the ability to model diel vertical migration (DVM) will be essential in many applications worldwide, especially large and deep lakes across the globe, as well as in many marine or estuarine environments.I In such cases the modeler will often have to rely on studies that specifically target zooplankton (or fish) DVM in similar environments. For fish however, a large step (if somewhat of a low-hanging fruit) towards more realistic model representation has been taken in WET, with the removal of the unrealistic absence of movement between model layers in its predecessor FABM-PCLake model.

While DVM in zooplankton and fish is an elusive dynamic to observe and understand, the importance of vertical movement processes for the composition and seasonal succession of phytoplankton communities is more easily recognized. Motile phytoplankton have a distinct advantage in highly stratified conditions, where the ability to stay in the euphotic epilimnion (Wentzky et al., 2020), or to balance opposing gradients of light and nutrients by migrating to the hypolimnion (Leach et al., 2018) can be important. We demonstrated that the choice of DVM method has profound impacts on model behavior (Fig. 5), e.g. whether mobile phytoplankton will concentration at the surface layer or near the bottom of the thermocline (i.e. forming a DCM).

The option of nitrogen fixation in phytoplankton is another feature which might improve model performance in stratified or oligotrophic lakes, where nitrogen limitation can be important (Reinl et al., 2021), although the ability of N-fixation to fully compensate for nitrogen limitation has recently been called into question (Shatwell and Köhler, 2019). In the present study, the impact of nitrogen fixation was expected to be low, as Lake Bryrup is predominantly limited by phosphorous, particularly in the main part of the growing season. Nevertheless, by adding a nitrogen-fixing cyanobacterial competitor, we shoved how shifting nutrient conditions throughout the growth season shaped the relative competitive landscape, as well as the nitrogen fixation rate of the N-fixing cyanobacteria throughout the model period.

**5.2 Advantages of model modularization** As demonstrated here, the WET model can reproduce the behavior of FABM-PCLake. But this food web configuration might not be optimal for every, nor even necessarily in this, use case. While beyond the scope of this paper, the WET model application to Lake Bryrup could potentially be calibrated in several food web configurations to find the optimum conceptual representation. This procedure could in principle be applied in all model applications as an extra layer in the calibration process, but will not be feasible in many cases, in the face of the already daunting task of calibrating a model with hundreds of parameters. Unfortunately, we believe that this is a case where there is no replacement for experience with both models and the study system, and the scientist will have to rely on their expertise to configure the most accurate and realistic food web for any given research question. However, on a smaller scale, it will often

be possible to test different food web compositions, by e.g. adding or subtracting an extra phytoplankton, zooplankton or fish module to an already calibrated model, and doing simpler recalibrations.

Differences in ecosystem structure and functions between lakes in different climatic regions would also most likely warrant changes to food web configurations. For example, fish populations in (sub)tropical shallow lakes have in general smaller body size, shorter life span, faster growth, multiple reproduction events and stronger preference for the littoral zone compared to temperate lakes (Meerhoff et al., 2012, and references herein). In combination with increased proportion of herbivorous and omnivorous fish species (González-Bergonzoni et al., 2012; Iglesias et al., 2017), this difference would most likely weaken trophic cascades and hereby diminish the impact of several lake restoration strategies (Jeppesen et al., 2010). So to reproduce for instance warm water shallow lake dynamics and responses to potential restoration efforts, configuration of the food web to the specific lake is likely needed.

## 5.3 Concluding remarks regarding the new features

Overall, the new changes that separate WET from its predecessors provide the model with a high degree of flexibility and adaptability, with the distinct advantage of allowing one model code base to handle many different application cases, instead of requiring many distinct models for different purposes. By taking advantage of the modularization, distinct food web configurations can be set up for different systems. Meanwhile, the new vertical movement and nitrogen fixation algorithms allow the model to be applicable in a much wider array of physical settings, across gradients in latitude, depth or salinity. This flexibility of the new model may contribute to limiting cases of parallel development and 'reinventing the wheel', while promoting comparability between different model implementations. As stated in the introduction, we believe that the consolidation of various model features and approaches into a few flexible and customizable models is a crucial process for the overall progress of the field of ecological modelling. However, for such models to be truly and successfully flexible, such customization must be possible with relative ease, or risk going unused. The modularized structure of WET under the FABM framework supports the user by making everything configurable in a single setup file, by making it easy to switch modules on or off, and by having all options relevant to an organism type available from the base module. By switching out module sections or an entire configuration file, a new model setup can be run with the same executable, without having to change or recompile any source code. For an even simpler workflow, running the model through the QWET plugin for the QGIS software provides the user with a simple step-by-step GUI based process, for which a tutorial is available on the WET homepage.

## 5.4 Future work

WET is under active and continuous development. Currently, effort is being applied to improve the applicability of the model to subtropical and tropical regions, which will include improvements to the macrophyte module and support for herbivory in fish. Another work in progress is a complete overhaul of the heterotrophic modules, replacing old feeding formulations with

more realistic descriptions, and introducing options for dynamic diets, feeding strategies and foraging efforts of fish, based on optimal foraging theory.

Apart from these new features, other areas for future work include improved handling of resuspension of sediment, a fully size-based fish module, and extensive testing and improvements of the model in a 3D application, with the expressed aim of the authors (in their roles as the current main developers of WET), that the model be even more tailorable to a wide array of ecosystems types, across latitudinal, spatial and productivity gradients, simply by turning features on or off, and combining different modules, all configurable at runtime. With this model, which is open source and freely available, we hope to facilitate the consolidation of successful features of many models together in one, with the goal of preventing 're-inventions of the wheel' in the future, and making aquatic ecosystem modelling easier, more flexible and, ultimately, better.

## 6 Code Availability

Name of software: WET (Water Ecosystems Tool) – version 1.0.

Developers: Dennis Trolle, Fenjuan Hu, Nicolas Azaña Schnedler-Meyer, Tobias Kuhlmann Andersen, Karsten Bolding & Anders Nielsen

Contact Address: Department of Bioscience, Aarhus University. Vejlsøvej 25, 8600 Silkeborg, Denmark

Email: wet.info@wet.au.dk

Availability: freely available under the GNU General Public License (GPL) version 2. Further information, executables and source code available at http://wet.au.dk, https://gitlab.com/WET or https://doi.org/10.5281/zenodo.6482852. A guide for model compilation, setup and configuration is available at the WET website (see 'For developers'). Follow this guide in order to download and compile the model.

## 7 Author Contributions

The model was coded and developed mainly by NASM, with significant contributions and aid from DT, TKA, FRSH and KB, based on a previous model code by FRSH. The testcase was adapted and carried out by TKA, and NASM prepared the manuscript with contributions from all co-authors.

## 8 Competing Interests

The authors declare that they have no conflict of interest.

## 9 Acknowledgements

Funding: This development has been supported by the CASHFISH project funded by the Danish Council for Independent Research, the WATExR project, funded through the EU JPI Climate initiative, the PROGNOS project, funded through the EU JPI Water initiative, a project on Mechanistic Models for Water Action Planning, funded by the Danish EPA, and through the Lake Stewardship Project, supported by the Poul Due Jensen Foundation. T.K. Andersen was supported by Ph.D. funding from the Sino-Danish Center for Education and Research.

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

**Table 1. Parameters and settings related to the new features in WET.**

| Module & feature | Parameter (prerequisite) | Explanation | Options or default value | Units | Note |
|---|---|---|---|---|---|
| **Phytoplankton** | | | | | |
| Modularization | lSi | toggle Si usage | .true. or .false. (default) | - | see Janse (2005) for explanation of other parameters related to silica uptake |
| N fixation | lNfix | toggle N fixation | .true. or .false. (default) | - | |
| | cNFixMax (lNfix = .true.) | maximum N fixation rate | 0.01 | mg N mg$^{-1}$ DW d$^{-1}$ | |
| | fMuNFix l (Nfix = .true.) | fraction of growth realized at max. N fix. rate | 0.9 | - | |
| Vertical migration | qTrans | vertical movement option | 1 = passive advection & buoyancy (default). 2 = nutrient taxis. 3 = light taxis. 4 = nutrient and light | - | |
| | cVSet (qTrans = 1) | settling or flotation rate | 0.05 | m d$^{-1}$ | negative in case of sinking |
| | cVswim (qTrans > 1) | vertical swimming speed | 10.0 | m d$^{-1}$ | |
| | fLVMmin (qTrans > 2) | minimum detectable PAR level | 0.025 | W m$^{-2}$ | |
| | fNutLimVMdown (qTrans = 2 or 4) | nutrient limitation triggering downwards taxis | 0.675 | - | |
| | fNutLimVMup (qTrans = 2 or 4) | nutrient limitation triggering upwards taxis | 0.75 | - | |
| **Macrophytes** | | | | | |
| Modularization | nCompts | number of macrophyte competitors | 0 | - | all competitor instances must be pointed to in the 'coupling' section |
| **Zoobenthos** | | | | | |
| Modularization | nprey | number of prey modules | 1 | - | all prey instances must be pointed to in the 'coupling' section |
| | cPref1 (nprey > 0) | selection factor for prey 1 | 1.0 | - | numbered copies for each prey item up to nPrey must be specified |
| | lSi1 (lSi of prey 1 must be .true.) | toggle prey silica tracking (diatom prey) | .true. or .false. (default) | - | optional. Allows calculation of Si excretion |
| | prey_suffix1 (nprey > 0) | add suffix to prey state variable coupling | character string (default is empty) | - | optional. Allows coupling to state variables that has a suffix after their standard state variable names |
| | lEatPOM | toggle POM consumption | .true. (default) or .false. | - | |
| | POMpref (lEatPOM = .true.) | selection factor for POM consumption | 1.0 | - | |
| **Zooplankton** | | | | | |
| Modularization | nprey | number of prey modules | 1 | - | all prey instances must be pointed to in the 'coupling' section |
| | cPref1 (nprey > 0) | selection factor for prey 1 | 1.0 | - | numbered copies for each prey item up to nPrey must be specified |
| | lSi1 (lSi of prey 1 must be .true.) | toggle prey silica tracking (diatom prey) | .true. or .false. (default) | - | optional. Allows calculation of Si excretion |
| | prey_suffix1 (nprey > 0) | add suffix to prey state variable coupling | character string (default is empty) | - | optional. Allows coupling to state variables that has a suffix after their standard state variable names |
| | lEatPOM | toggle POM consumption | .true. (default) or .false. | - | |
| | POMpref (lEatPOM = .true.) | selection factor for POM consumption | 1.0 | - | |
| Vertical migration | qTrans | vertical movement option | 0 = no transport. 1 = passive transport. 2 = hypoxia avoidance (default). 3 = hypoxia avoidance and light-based movement | - | |

| | | | | | |
|---|---|---|---|---|---|
| | Vswim (qTrans > 1) | vertical movement speed | 15.0 | m d$^{-1}$ | |
| | cMinO2 (qTrans > 1) | oxygen concentration limit | 2.0 | mg O$_2$ L$^{-1}$ | |
| | cMinLight (qTrans = 3) | light level triggering upwards swimming | 40.0 | W m$^{-2}$ | |
| | cMaxLight (qTrans = 3) | light level triggering downwards swimming | 40.0 | W m$^{-2}$ | |

**Fish**

| | | | | | |
|---|---|---|---|---|---|
| Modularization | qStageOpt | toggle stage coupling | 0 = no coupling (default). 1 = reproduction. 2 = maturation. 3 = reproduction and maturation | - | for qStageOpt > 0, all coupled instances must be pointed to in the 'coupling' section |
| | cDayReprFish (qStageOpt = 1 or 3) | reproduction date | 120.0 | day-of-year | |
| | fReprFish (qStageOpt = 1 or 3) | yearly reproduction fraction | 0.02 | y$^{-1}$ | total fraction of biomass transferred to coupled instance |
| | cDayAgeFish (qStageOpt = 2 or 3) | reproduction date | 360.0 | day-of-year | |
| | fAgeFish (qStageOpt = 2 or 3) | yearly aging fraction | 0.5 | y$^{-1}$ | total fraction of biomass transferred to coupled instance |
| | nKShares (qStageOpt > 0) | toggle shared carrying capacity with other instances | 0 | - | optional. All instances sharing the carrying capacity must be coupled through the coupling section |
| | fFishZoo | zooplanktivory effort fraction | 1.0 | - | fFishZoo, fFishBen and fFishPisc must sum up to one, otherwise WET will force them to |
| | fFishBen | zoobentivory effort fraction | 0.0 | - | fFishZoo, fFishBen and fFishPisc must sum up to one, otherwise WET will force them to |
| | fFishPisc | piscivory effort fraction | 0.0 | - | fFishZoo, fFishBen and fFishPisc must sum up to one, otherwise WET will force them to |
| | nZOO | number of zooplankton prey | 0 | - | all prey instances must be pointed to in the 'coupling' section |
| | ZOOpref1 (nZOO > 0) | preference factor for zooplankton prey 1 | 1.0 | - | numbered copies for each prey item up to nZOO must be specified |
| | nBEN | number of zoobenthos prey | 0 | - | all prey instances must be pointed to in the 'coupling' section |
| | BENpref1 (nBEN > 0) | preference factor for zoobenthos prey 1 | 1.0 | - | numbered copies for each prey item up to nBEN must be specified |
| | kTurbFish (nBEN > 0) | relative resuspension by fish bottom feeding | 1.0 | g g$^{-1}$ DW d$^{-1}$ | |
| | nPISC | number of fish prey | 0 | - | all prey instances must be pointed to in the 'coupling' section |
| | PISCpref1 (nPISC > 0) | preference factor for fish prey 1 | 1.0 | | numbered copies for each prey item up to nBEN must be specified |
| | lVegOpt | toggle for macrophyte dependent fish growth | .true. or .false. (default) | - | mainly for backwards compatibility with the PCLake model family; see Janse (2005) |
| Vertical migration | qTrans | vertical movement option | 0 = no transport. 1 = passive transport. 2 = hypoxia avoidance (default). 3 = hypoxia avoidance and light-based movement | - | |
| | Vswim (qTrans > 1) | vertical movement speed | 10.0 | m d$^{-1}$ | |
| | cMinO2 (qTrans > 1) | oxygen concentration limit | 2.0 | mg O$_2$ L$^{-1}$ | |
| | cMinLight (qTrans = 3) | light level triggering upwards swimming | 1.0 | W m$^{-2}$ | |
| | cMaxLight (qTrans = 3) | light level triggering downwards swimming | 40.0 | W m$^{-2}$ | |

**Table 2.Comparison of performance metrics between Chen et al. (2020) and the present study.**

| WET (this study) | | | | | Chen et al. 2020 | | | |
|---|---|---|---|---|---|---|---|---|
| $R^2$ | | | RMSE | | $R^2$ | | RMSE | |
| Variable : | calibration | validation | calibration | validation | calibration | validation | calibration | validation |
| Temp. | 0.98 | 0.98 | 1.19 | 1.27 | 0.98 | 0.98 | 1.37 | 1.41 |
| DO | 0.64 | 0.56 | 2.4 | 2.66 | 0.45 | 0.38 | 3.09 | 3.15 |
| NO3 | 0.8 | 0.75 | 1.04 | 1.27 | 0.85 | 0.85 | 0.69 | 0.71 |
| NH4 | 0.69 | 0.39 | 0.2 | 0.2 | 0.45 | 0.51 | 0.32 | 0.15 |
| TN | 0.7 | 0.65 | 1.23 | 1.5 | 0.79 | 0.81 | 0.71 | 0.78 |
| PO4 | 0.61 | 0.26 | 0.02 | 0.03 | 0.3 | 0.19 | 0.03 | 0.04 |
| TP | 0.58 | 0.23 | 0.07 | 0.08 | 0.31 | 0.15 | 0.08 | 0.08 |
| Chl. a | 0.26 | 0.39 | 24.49 | 30.61 | 0.29 | 0.28 | 23.65 | 35.91 |

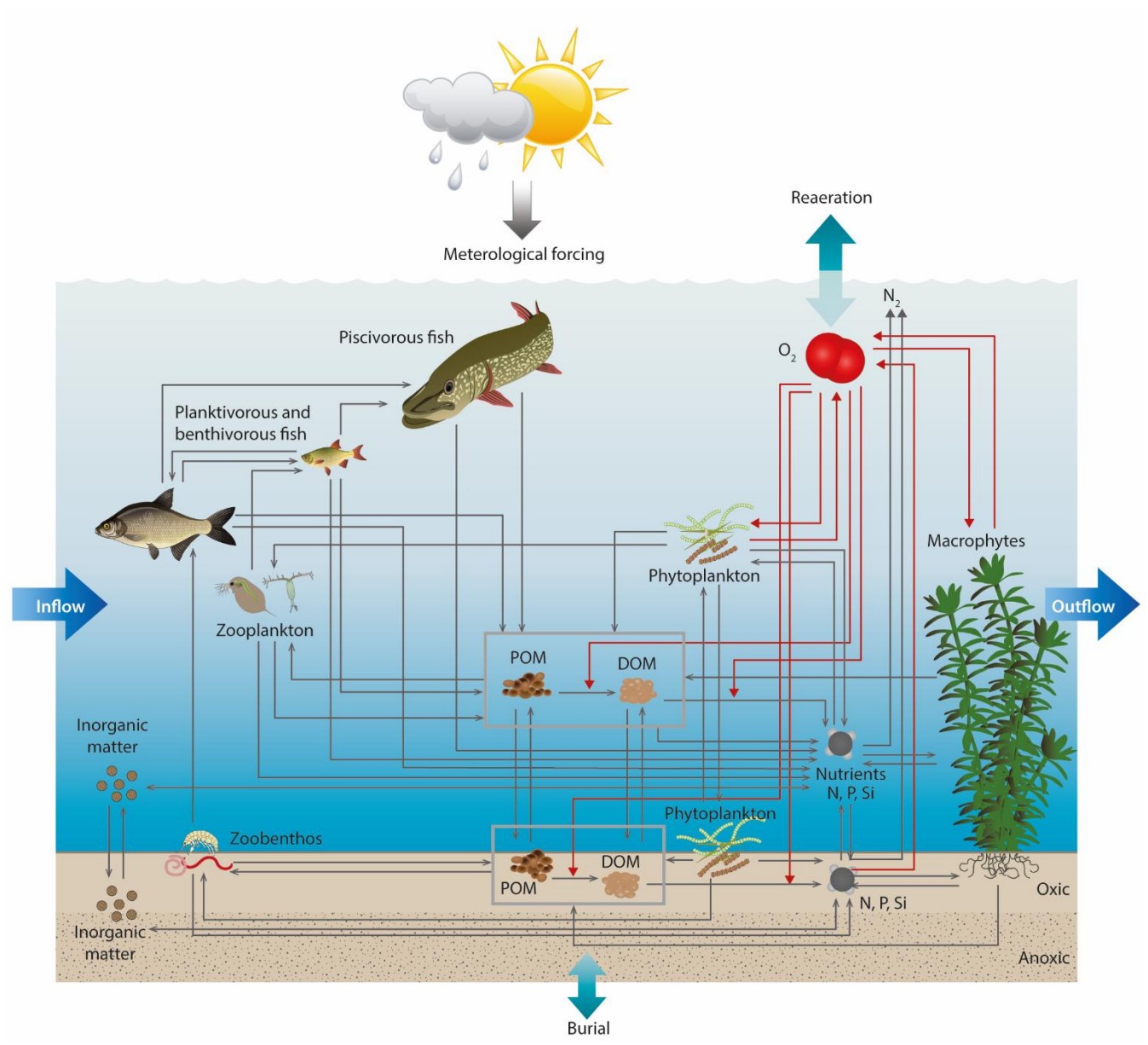


**Figure 1. The conceptual model of WET in a setup similar to the ones of 0D PCLake and 1D GOTM-FABM-PCLake. Grey arrows indicate fluxes of matter (dry-weight, nitrogen, phosphorous or silica) between ecosystem pools and red arrows indicate oxygen uptake/production. Adapted from Hu et al. (2016).**

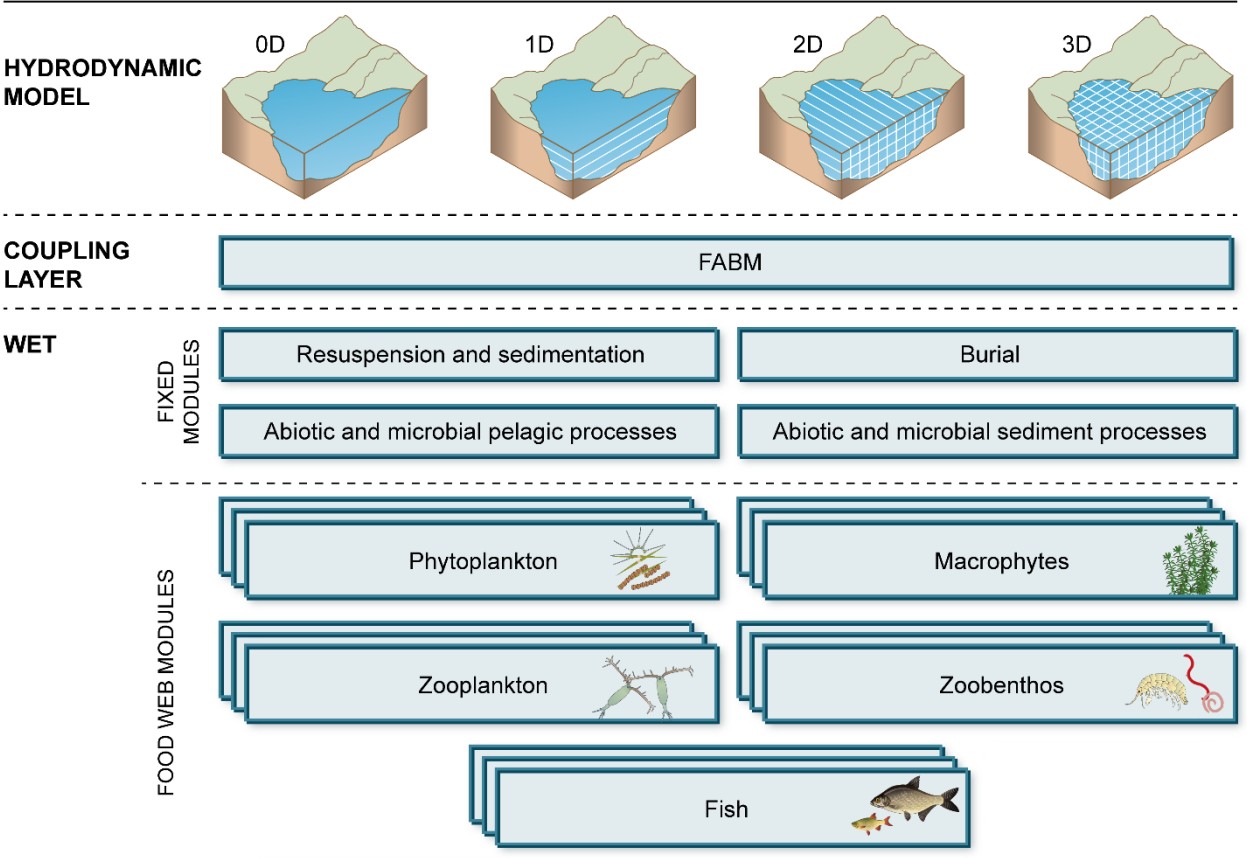

**Figure 2. Functional structure of a WET setup. The ecological model WET is coupled to a physical driver model (of any dimensionality) through the coupling interface FABM. WET is partitioned into a number of fixed modules, handling microbial and chemical processes, and a flexible set of fully modularized food web modules, which can be duplicated and combined as required.**

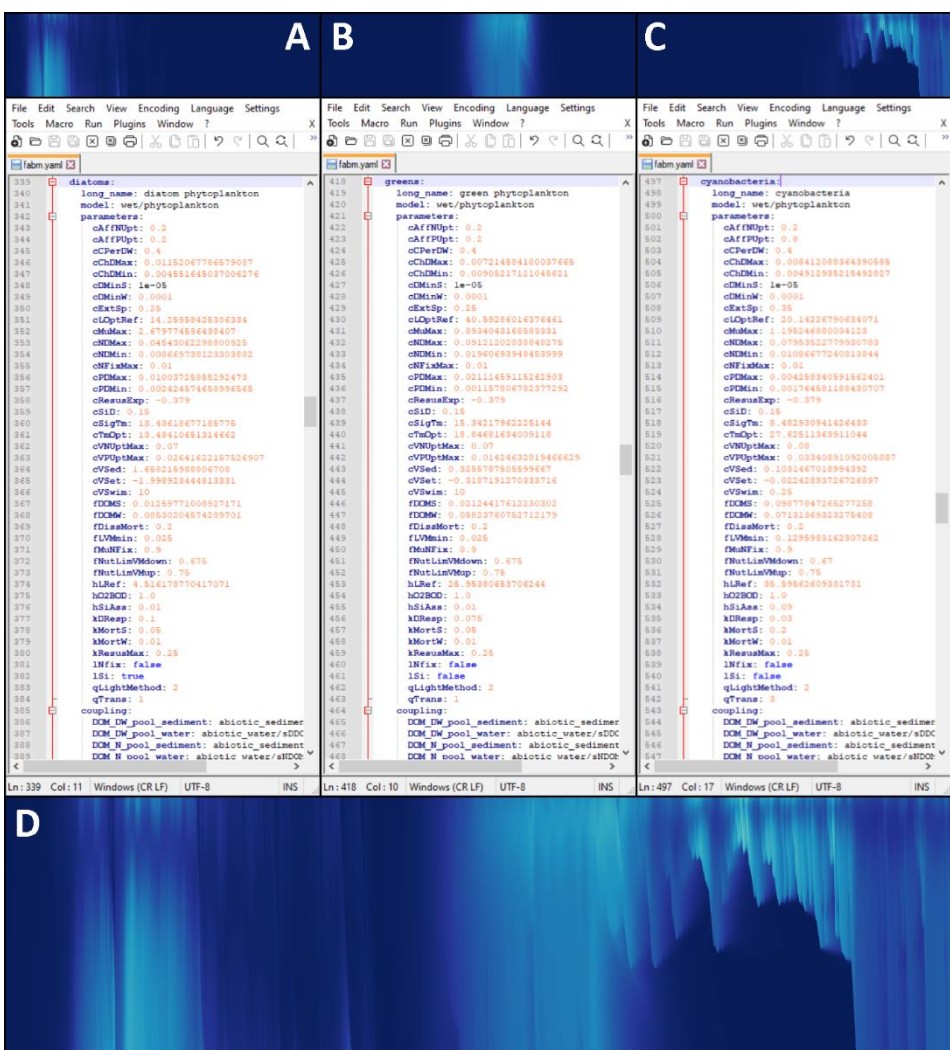

Figure 3: Modular configuration of phytoplankton instances in WET. Top row: Individual chlorophyll a concentrations of diatoms (A), green algae (B) and motile cyanobacteria (C). Middle row: view of the section of the model configuration file corresponding to the panel above. Bottom panel (D): Total chlorophyll a concentration for the upper 6 meters of lake Bryrup for the year 1994, as simulated by WET coupled to the 1D GOTM lake model. Panel D identical to of Fig. 5E, and all axes and colormaps of panels A-C likewise identical to those of Fig. E, except the shown period in this figure is first of April through the first of September.

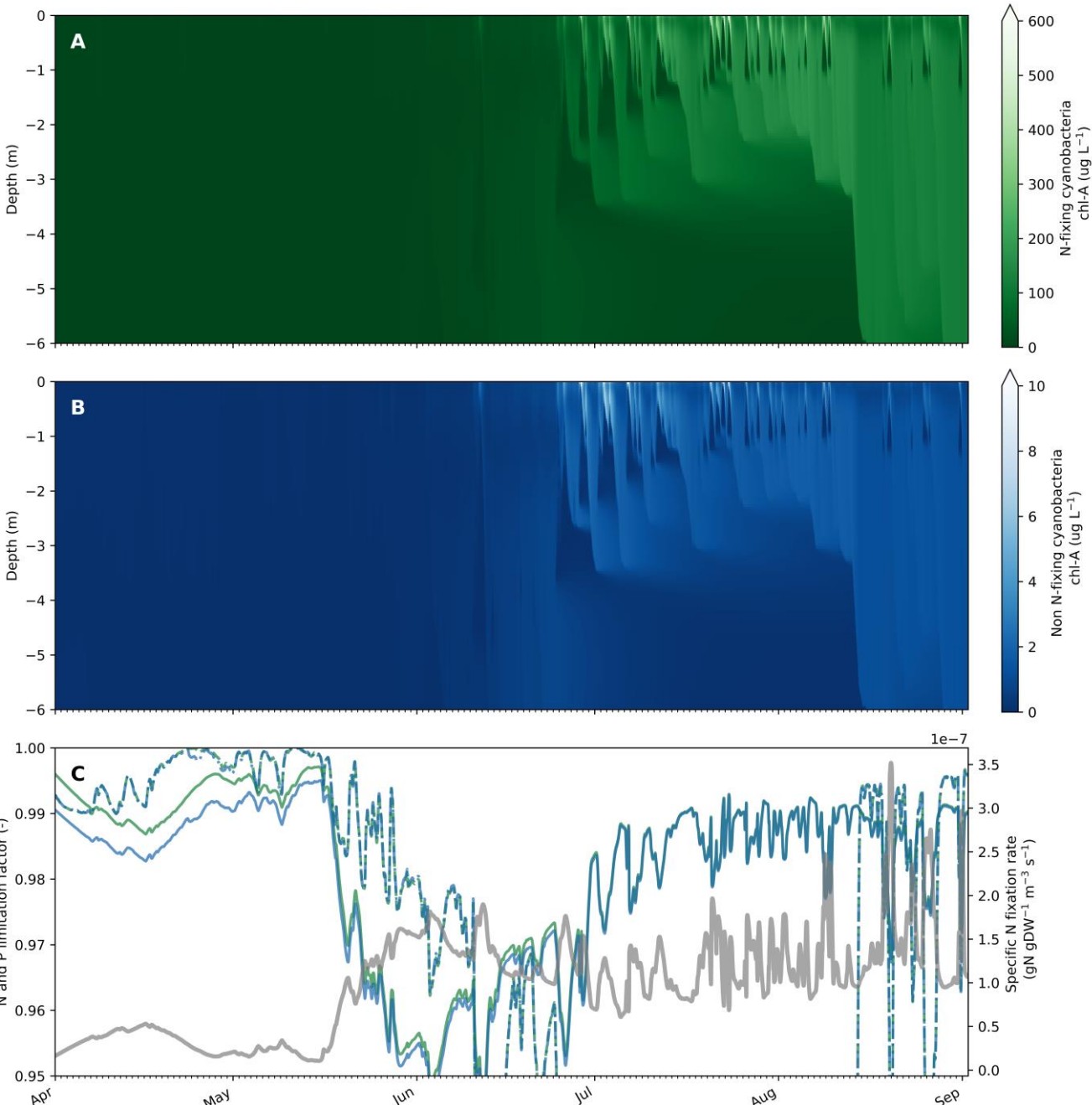


**Figure 4: Example of simulated cyanobacteria, nutrient limitation and N fixation dynamics in Lake Bryrup from April to September 1994. Output of N-fixing and non N-fixing cyanobacteria chl-A concentrations from the upper 6 m of Lake Bryrup (*A* and *B*). C: N and P limitation factor (solid and dashed lines, respectively) for N-fixing and non N-fixing cyanobacteria (green and blue lines, respectively) and specific N fixation rate by N-fixing cyanobacteria (grey line, secondary y-axis) in surface model layer (surface to 665 0.5 m). The phytoplankton group is nutrient (N and/or P) replete or deplete when the limitation factor is equal to 1 or 0, respectively.**

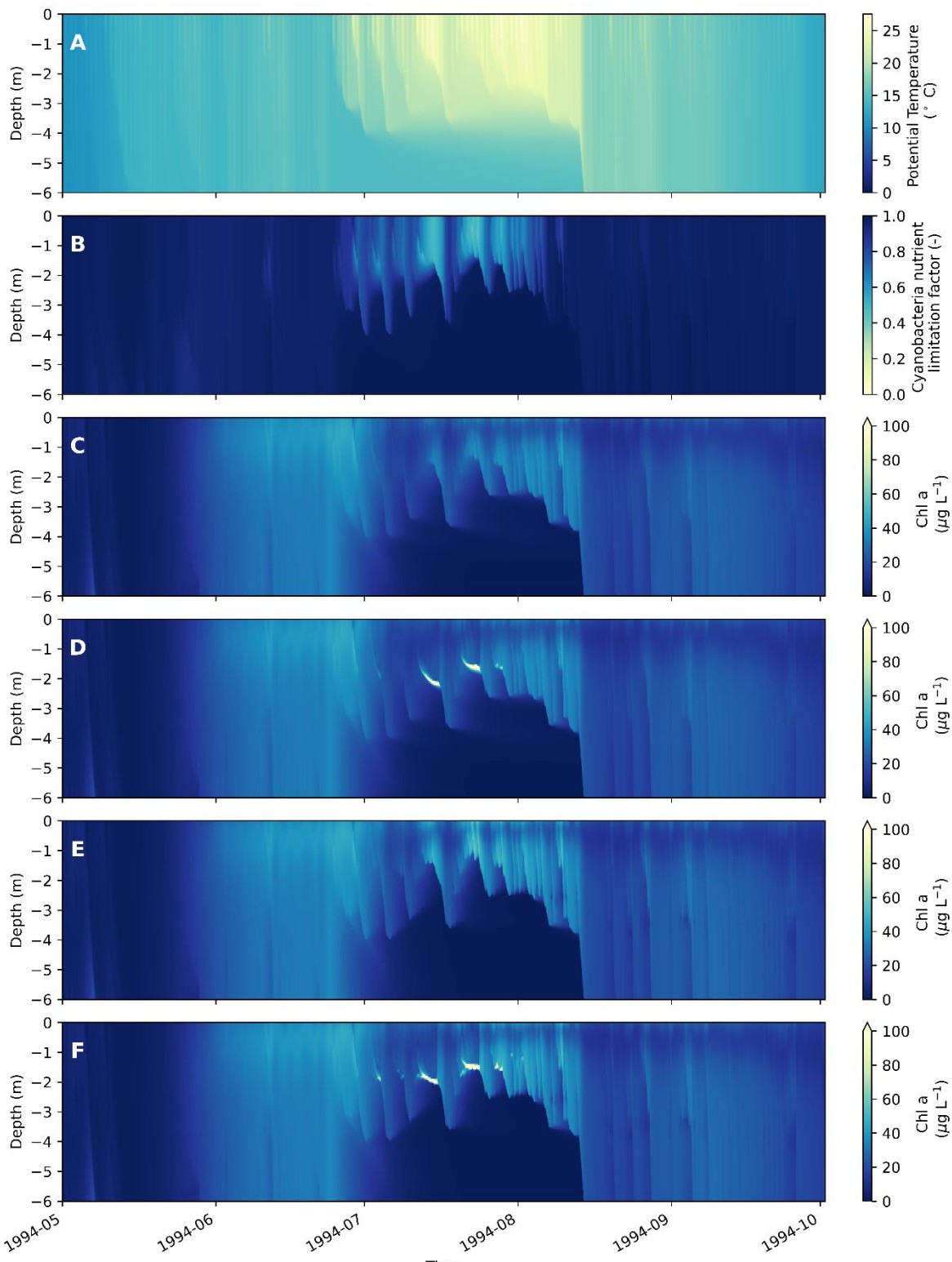

**Figure 5: Output from the upper 6 meters of Lake Bryrup for the year 1994, as simulated by WET coupled to the 1D GOTM lake model. A: Water temperature. B: Overall nutrient limitation factor for the cyanobacteria instance. C-F: Total water column chlorophyll a, for different settings of cyanobacterial vertical movement. C: passive movement (qTrans = 1). D: nutrient taxis (qTrans = 2). E: Phototaxis (qTrans = 2). F: combined photo- and nutrient taxis. For all panels, the cyanobacterial swimming speed, cVSwim was set to 0.25 m d$^{-1}$. The other vertical movement paraters were set to: fNutLimVMdown = 0.67, fNutLimVMup = 0.75, and fLVMmin = 0.13 W m$^2$.. For C-F, note fully mixed diatom bloom in spring, followed by a summer bloom of cyanobacteria in late summer. For legibility of the figure, the maximum colorbar cutoff for C-F has been set at a lover value than brief and localized very high chlorophyll a concentrations (up to ~1300 µg L$^{-1}$ in F).**

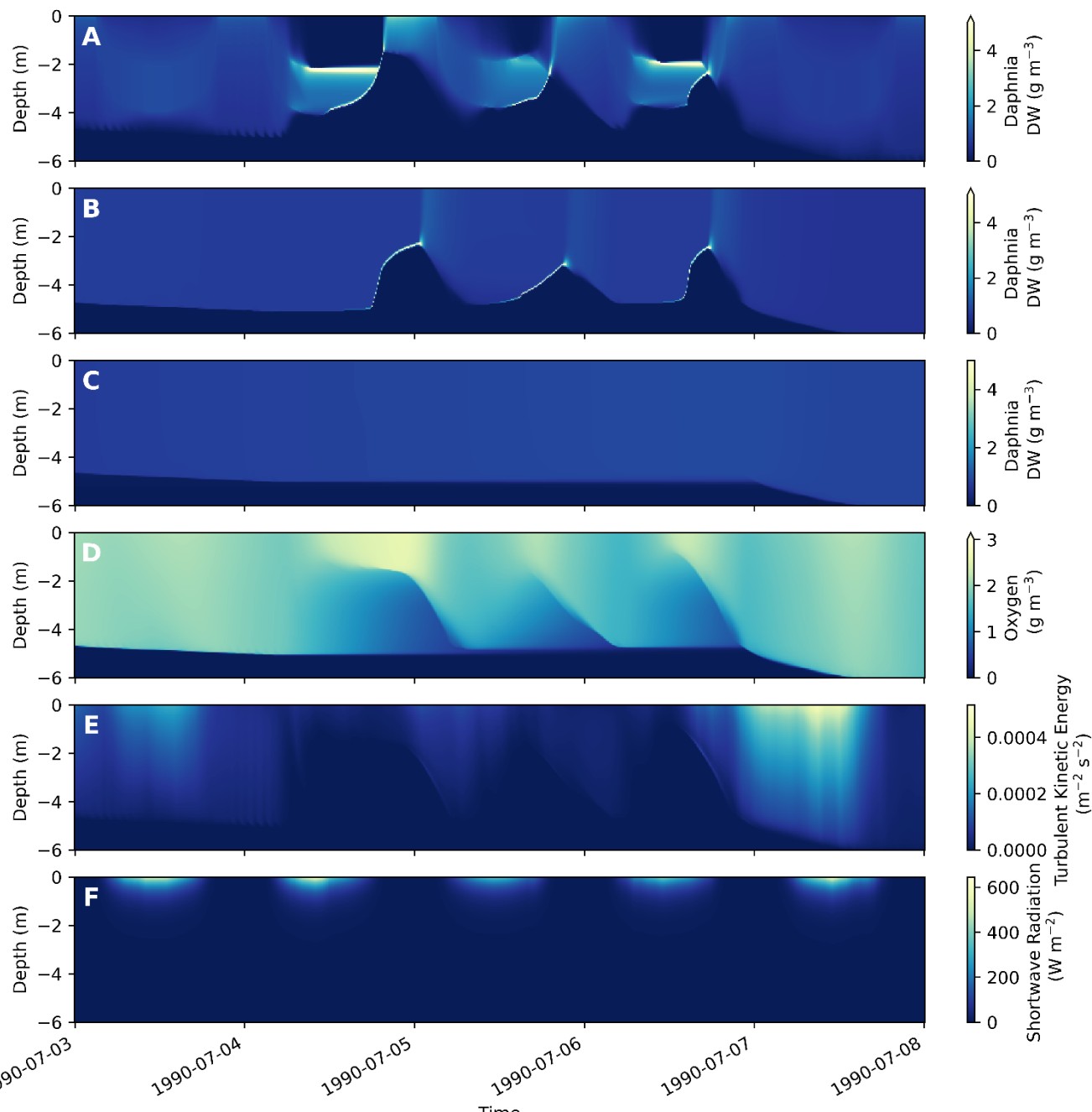

**Figure 6. Example of simulated vertical migration in WET.** Output from the upper 6 meters of Lake Bryrup during five days in July 1994, as simulated by WET coupled to the 1D GOTM lake model. A: Vertical distribution of daphnia dry-weight biomass, with vertical movement set to lightbased vertical movement. Note vertical movements of zooplankton biomass, triggered by light intensity in the water column, as well as oxygen depletion in the hypolimnion, and modulated by turbulent mixing. B: Vertical distribution of daphnia dry-weight biomass, with vertical movement set to hypoxia avoidance. C: Vertical distribution of Daphnia dry-weight biomass, with vertical movement set to passive transport only. D-F: Oxygen, Turbulent, and Light conditions, respectively, which informs the vertical movement of zooplankton. For legibility of the figure, the maximum colorbar cutoff has been set at a lover value

685     **than brief and localized very high daphnia concentrations (~30 g m-3) during the anoxia avoidance episodes. B: Downwelling shortwave radiation. For these simulations, daphnia vertical movement parameters were set to: Vswim = 45 m d$^{-1}$, cMaxLight = 40.0 W m$^{-2}$, cMinLight = 0.15 W m$^{-2}$, and cMinO2 = 2.0 g O2 m$^{-3}$.**