# Peer review of "Water Ecosystems Tool (WET) 1.0 - a new generation of flexible aquatic ecosystem model"

_Geoscientific Model Development, 2021_

## Author Response (AR1)

Dear Editor,

Thank you again for our considering our manuscript suitable for publication in GMD. We received two great reviews, which pointed out several areas where our manuscript could be improved. We have redone most figures, as well as revised the manuscript thoroughly in line with the reviewer's suggestions. Please find below detailed answers (in black text) to the reviewers' comments (in blue). We believe that the manuscript has greatly improved with this revision and hope that it is now suitable for publication.

Sincerely,

Nicolas Azaña Schnedler-Meyer, on behalf of the authors.

**Referee #1:**

First comment:

First, to make the tool useful for broad users, it should be easy to deploy and configure. However, for many open-source tools, the developers focus on more on development but documentation. It is understandable for those new developments. But for a tool that is built on a mature model framework, I did not expect it. I downloaded the source code and test case from the provided link. However, there are none of documentations in the folder that teach the users how to compile and configure. I typed "make" in the folder but cannot compile WET successfully. For most of the users, they will give up after this first attempt. It is unfortunate for a tool that can benefit the society.

While we realize that this was not clearly stated anywhere in the manuscript, comprehensive documentation on how to download and compile WET is available from the WET website at wet.au.dk under 'for developers'. We have highlighted this fact in the code availability section of the manuscript, and will refer to this guide in the README.txt file as well. Regarding model usability, we have revised the description of QWET (i.e. the graphical user interface for the GOTM-WET model complex) to reflect that a tool is available for model users to ease model configuration and execution (lines 76-82).

Second comment:

Second, the authors stated that WET can be used to test for the optimal food web configuration in a specific case. But they have very limited discussions about it. For example, how should we distinguish the effect of model calibration and module settings in

the situation that both can improve the model performance? As a complex model presented here, there may be tens of different modules. Each module may have more than ten parameters. Under such complex situations, the optimal food web configuration is easily said by done. So I want to hear some insights from the authors.

We might have presented the wrong impression here, by implying that there is a tested workflow for doing this. Unfortunately, this is indeed an area where the researcher must make use of their own judgement. We tried to allude to these thoughts in the original section 5, lines 354-358. We have changed lines 112-113 to be more precise, and improved on the discussion of this aspect in section 5.2.

Third comment:

Third, does WET have any unit testing features? As the tool is extensively modulized and supposed to be under community development, unit testing would be a key procesure to ensure software quality.

There is as of yet no comprehensive unit testing suite available for WET. WET is currently maintained and developed by a small team of researchers, and our resources do unfortunately not cover this. Thus, users must be prepared to check their results for unexpected or erroneous behaviors, and are encouraged to post any concerns as e.g. support questions on the gitlab website. However, as WET is built for the FABM framework, it does benefit from the excellent error handling within FABM. In addition, some tools for e.g. stress-testing models are available for the FABM framework, but their description is unfortunately beyond the scope of this manuscript.

Reviewer's comment:

L268: what does "Each water layer included a sediment layer of 10 cm" mean? Please illustrate.

In this case, this is a feature of the lake GOTM's hypsograph setup. In order to capture lake sediment-water column interactions at all depths, the bottom is effectively split up, such that each model layer in the 1D setup has an attached bottom layer. Interactions between the water column of a layer, its attached bottom, and the water column layer below is governed by the hypsograph, which specifies the fraction of the bottom area to total layer area. We have added a description of this setup to section 3.1 and have included a reference, which contains a further description of the hypsograph.

We would however like to avoid tying the mind of the reader too tightly to the chosen physical model used, as WET can in principle be utilized with a variety of physical models, and so we would prefer to avoid a figure of the hypsograph setup in this manuscript.

Reviewer's comment:

L279: Are boundary conditions only set at surface layer or all water layers? If the latter, how the inflow are distributed across different layers?

We have specified that both in- and outflow was applied in the top layer (line 309).

Reviewer's comment:

L284: In Chen et al. (2020), ACPy was said to be used for calibration. What is the relationship between ACPy and parsac?

ACPy is identical to the Parsac package, except for the name, which has been changed in the meantime.

Reviewer's comment:

L306: runs at a lower resolution?

We are unsure what the reviewer mean here, but to clarify, we ran the model at higher resolution, in order to increase detail on e.g. the vertical movement dynamics. We have clarified some aspects of this in section 3.1

Reviewer's comment:

L336: Section 5 needs to be divided into several sub-sections, for example model performance, model limitation and future work.

We have separated this section into four sub-sections.

**Referee #2:**

First major comment:
1. My major concern is a lack of sufficient validation for the new functions to the model in the current manuscript. Clearly these new features, such as vertical migration and N fixation of cyanobacteria, are crucial for modeling the food web interaction and ecosystem dynamics, which jointly improve the applicability of the WET model. However, the test case of Lake Bryrup and the available dataset are insufficient to explicitly test the adequacy of the new functions. The lake is also a bit too shallow to reflect the features of the deep lakes with pronounced vertical migration of organisms. More importantly, the main advance of the 'flexible food web' is also not

addressed in the test case. Nevertheless, I can understand that field data to validate these new features are rare, particularly the spatial distributions of the organism along the vertical dimension. Therefore, I recommend to use the case of Lake Bryrup, but rather than validating the model, the main focus here should be on providing a more detailed evaluation of the new features. In additional to the existing analyses, the authors may consider the following (but not limited to): 1) compare the modeled cyanobacteria with and without N fixing function (ideally with two types of cyanobacteria), and illustrate how the N limitation regulates the growth of the cyanobacteria and potentially other algae groups; 2) design a gradient of the vertical migration parameters for zooplankton, as shown currently in Fig. 5 as an nice example (but now only with one parameter set). Such additional evaluations will help readers to better understand the importance of the new features for modeling and unravel the sensitivity of the key parameters, thereby opening up the opportunities to test the new model in other lake cases with more comprehensive datasets in the future.

We appreciate that the reviewer recognizes the challenges involved, but agree that the model tests can be improved. We are thankful for the excellent suggestions and recommendations on how to improve this aspect of the manuscript. We have gone with the suggestion of the reviewer and provided a more thorough presentation of the new features, in accordance with suggestion 1) and 2) of the reviewer's comment.

1) As suggested by the reviewer, we have added a second, N-fixing cyanobacteria to the model, and compare the dynamics of these in Lake Bryrup in a new figure (Fig. 4). As Lake Bryrup is mainly P-limited during the growth season (as are most temperate lakes), we have also run scenarios with N-inflow to the lake disabled, and report and discus the dynamics observed in a new section (4.2), and in the discussion (5.1).

2) We interpret this suggestion to mean that the reviewer would like us to compare the dynamics of the different vertical migration options in the model. To this end, we have introduced three new panels in figure 5 (now 6), illustrating the dynamics of passive transport, and hypoxia avoidance by zooplankton, and comment on the results in the results and discussion.

Second major comment:
2. I found that some details of the new features are not clearly described. This could be fine for current users but may puzzle the new users in the future, thus obscure the application and further development of the model. The authors may consider using more Tables to list the new features of some modules with the explanation of the related parameters and meaning of the values in the configuration files. Below I list some specific points in the minor comments, but overall I would appreciate taking advantages of Tables in such model description papers as the key document.

We have done as suggested, and added a table (new Table 1) detailing the parameters and options related to each new feature. We refer to this table in the text, and have consequently

taken out many parameter names from the main text, where they disrupted the flow of the text, as also pointed out by the review below.

3. It remains unclear how to practically change the structure of the model by adding or deleting organism components. Likely it will be done in the configuration file, but it is not described (if I did not miss anything). Tutorial cases and/or a detailed manual for such configuration procedure should be provided because such flexible feature is one of the major advantages of the new WET over the previous version and also many other aquatic ecosystem models. Besides, I recommend to briefly discuss the possibility to implement the flexible food web in a more user-friendly way (such as the GUI of ECOPATH or AQUATOX), if such function would be envisioned for further model development.

This comment is in line with reviewer #1, who also requested more documentation on model setup and configuration. Although we failed to adequately cover this in the manuscript, there is in fact extensive documentation on how to setup and configure the model on the WET homepage. We have emphazised this more clearly in the manuscript. We have also included a short description on how to add or remove food web modules to the manuscript (section 2.1). With regards to the possibility of running the model with a more user-friendly interface, the QWET plugin for the (freeware) GIS software QGIS does exactly this. There is also extensive documentation in the QWET section of the WET homepage on how to set up QWET. We have further emphasized these facts in the manuscript (lines 76-82, 463-464 and 489-490).

4. It is uncommon to number the first version of the WET by '0.1.0'. The first release of software or model is usually numbered as '1.0' or '1.00' etc. Current version of '0.1.0' gives the impression that the model is still at the very beginning stage and far from completed for first release. Please consider changing the version numbering, if appropriate.

In this case, we simply followed the examples of some other model software, but upon reflection, we agree with your comment. We have changed the references to the software version in the manuscript, and will make another release of the model under the version number 1.0, when and if our manuscript reaches the accepted status.

Below, we list our reply to those minor comments where we did not just do as suggested by the reviewer:

1. Line 84: I am not sure if all the biogeochemical processes equation are 'unchanged' in WET compared to PCLake. If so, it is contradictory to the previous statement that processes such as resuspension are changed compared to previous model. Better add 'mostly unchanged', if proper.

We believe that the reviewer has misread the text here, as we do not actually claim to have changed any processes not explicitly presented in the manuscript. As such, we believe this statement is justified as is.

1. Figure 2: suggest to change the subtitle in vertical 'FOOD WEB MODULES' to 'FLEXIBLE FOOD WEB MODULES' to highlight the features of the model.

We did not follow the reviewer in this, as we believe that the modularized nature of these modules is apparent from the chosen graphics, the figure text, and main text of the manuscript. However, if the editor deems it neccessary, we will of course redo this figure.